# Group Equivariance Meets Mechanistic Interpretability: Equivariant Sparse Autoencoders

**Ege Erdogan, Ana Lucic**
University of Amsterdam
{e.erdogan, a.lucic}@uva.nl

## Abstract

Sparse autoencoders (SAEs) have proven useful in disentangling the opaque activations of neural networks, primarily large language models, into sets of interpretable features. However, adapting them to domains beyond language, such as scientific data with group symmetries, introduces challenges that can hinder their effectiveness. We show that incorporating such group symmetries into the SAEs yields features more useful in downstream tasks. More specifically, we train autoencoders on synthetic images and find that a single matrix can explain how their activations transform as the images are rotated. Building on this, we develop *adaptively equivariant SAEs* that can adapt to the base model's level of equivariance. These adaptive SAEs discover features that lead to superior probing performance compared to regular SAEs, demonstrating the value of incorporating symmetries in mechanistic interpretability tools.

 https://github.com/ege-erdogan/equivariant-sae

## 1 Introduction

ML models are increasingly used for scientific problems, from higher-level data analysis and hypothesis generation [1, 2] to lower-level emulation of physical processes such as protein folding [3]. Being able to interpret the internals of these models would not only ensure they are scientifically reliable and controllable, but potentially lead to novel scientific discoveries. Scientific data is often characterized by underlying symmetries: transformations such as rotations or translations that alter particular attributes in particular ways, and accounting for those symmetries can lead to more data-efficient models [4]. While mechanistic interpretability methods, in particular sparse autoencoder (SAEs), have been applied to scientific ML models in domains such as proteins [5, 6, 7, 8] and cell images [9, 10], there has been no work on how input symmetries can aid mechanistic interpretability. In this paper, we present early results suggesting that building SAEs that automatically adapt to symmetries in the data can greatly improve their performance in downstream tasks.

A set of symmetries such as rotations can be modeled as a *group $G$*. Groups *act on* sets such as images; e.g. with $g \in G$, $g\mathbf{x}$ can denote the rotated version of an image $\mathbf{x}$. Transformations of the same element $\mathbf{x}$ form an *orbit* $\{g\mathbf{x} : g \in G\}$. We can then split the features of $\mathbf{x}$ into those that are **invariant** with respect to $G$ and those that are **equivariant**. For example, the types of atoms in a molecule would be features invariant under 3D rotations, but the force vectors acting on each atom would be equivariant features, rotating along with the molecule. More generally, we define invariant features as those shared across an orbit: the set of possible options that $\mathbf{x}$ can be transformed into, while equivariant features depend on the particular transformation applied to $\mathbf{x}$. We provide a more detailed background on SAEs and groups in Appendix B.

There are two main challenges when training SAEs on data with group symmetries. First, the optimally sparse solution learns one latent per transformation for each equivariant feature, requiring

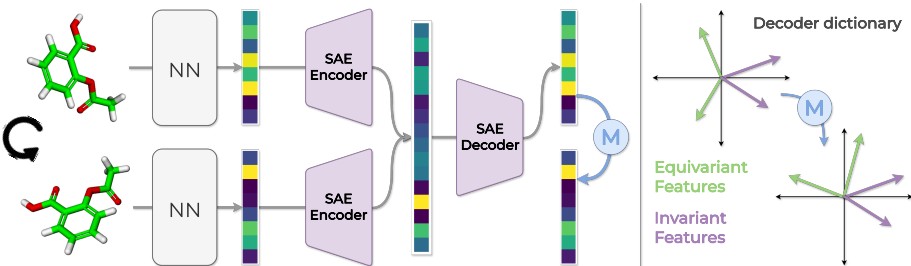

Figure 1: *Left:* We train an **invariant SAE** that maps activations of the transformed inputs to the same latents, and optimize the matrix $\mathbf{M}$ to estimate how the activations transform, achieving **equivariance**. *Right:* Transforming the decoder dictionary $\mathbf{D} \mapsto \mathbf{MD}$ allows us to observe which features discovered by the SAE are **invariant** or **equivariant** with respect to input transformations. We provide a more detailed figure of the mathematical operations involved in Figure 5 in the Appendix.

$O(|G|)$ latents per semantic feature, which is undesirable for larger groups. Second, we do not know *a priori* the degree of equivariance in base model activations, making it difficult to enforce exact symmetries in advance. In this work, we demonstrate that addressing these challenges can lead to improved performance in downstream applications. More specifically:

- We show that **a single matrix $\mathbf{M}$ acting on the activations explains more than 98% of the variance in the transformed activations** for MLP and CNN-based autoencoders trained on a synthetic image dataset transformed under the group of 90° rotations.

- We design **adaptively equivariant SAEs** consisting of an *invariant* autoencoder (avoiding the pitfall of exploding number of latents) which is made *equivariant* with the addition of $\mathbf{M}$ (avoiding the pitfall of unnecessarily-exact equivariance). This allows us adapt to what the model has learned rather than prematurely enforcing exact symmetries.

- We demonstrate **our adaptively equivariant SAEs learn more useful features that outperform regular SAEs in a set of binary probing tasks**, despite lagging behind in the reconstruction/sparsity frontier.

## 2 Related work

Our work is one of the first to bridge ideas from the sparse dictionary learning literature and the equivariant representation learning literature with a particular application towards mechanistic interpretability. **Learning approximate equivariance** via objectives similar to our SAE training objective has been proposed in [11, 12], and our general approach of learning an invariant encoder and a separate mapping from the canonical outputs to their original forms follows that of [13]. Our main difference is that we adapt these approaches to the space of neural network activations, where the symmetries are induced by the input transformations and are not well-defined.

Another line of related work concerns **learning symmetries and group representations** from data directly [14, 15]. Most similar to our approach of learning the transformation matrix $\mathbf{M}$, [15] proposes the MatrixNet architecture to learn matrix representation of group elements while staying faithful to the group axioms. Although it solves a similar problem, such an approach is not directly practical over neural network activations with dimensions on the order of $10^3$. Processing the resulting matrix representation ($O(10^6)$ parameters) with a neural network would lead to an explosion in the total number of parameters. That is why we instead opt for optimizing $\mathbf{M}$ directly.

To potentially avoid this explosion of the number of parameters, [14] propose an adversarial training method to learn potentially nonlinear symmetries by first mapping the observations to a latent space using an autoencoder and learning a linear group action in that latent space. While technically applicable to neural network activations, modeling the group action over a latent space further removed from the already difficult-to-interpret neural network activations could make interpretability, which is our main goal, more challenging.

Group-equivariant sparse dictionary learning methods have also been proposed [16, 17] although such exact symmetries cannot be enforced over neural network activations as we do not know to what degree they are equivariant. Finally, [18] proposes group crosscoders to analyze how the features learned by a neural network change as its inputs are transformed, constructing each dictionary vector with $G$ blocks each the size of the inputs. The size of the dictionary thus scales linearly with $|G|$ unlike our approach where the number of parameters is constant with respect to $|G|$.

## 3 Method: SAEs with adaptive equivariance

We consider groups $G$ where all transformations can be obtained as powers of a generator $g \in G$, i.e. $G = \{g^p\}_{p=1}^{|G|}$. Our design consists of a group-invariant TopK SAE [19, 20] with a two-layer encoder $E$ and linear decoder $D$, and a matrix $\mathbf{M}$ that learns to fit how the base model's activations transform as its inputs are transformed with the action of group $G$. Thus, with canonical (one representative per orbit) inputs $\mathbf{x} \in \mathcal{X} \subset \mathbb{R}^n$ and model activations $\psi(\mathbf{x}) \in \mathbb{R}^d$, for $p = 1, ..., |G|$,

$$D\left(E\left(\psi(g^p\mathbf{x})\right)\right) = \psi(\mathbf{x}), \quad \text{and} \quad \mathbf{M}^p D\left(E\left(\psi(g^p\mathbf{x})\right)\right) = \psi(g^p\mathbf{x}). \tag{1}$$

First, the SAE reconstructs all activations $\psi(g^p\mathbf{x})$ of the transformed inputs as the canonical activation $\psi(\mathbf{x})$. Then this reconstruction is transformed with $\mathbf{M}$ to obtain $\mathbf{M}^p\psi(\mathbf{x}) = \psi(g^p\mathbf{x})$.

**Invariant SAE.** We make our SAE invariant to group transformations of the base model's inputs by training it with the following *invariance loss*:

$$\mathcal{L}_{\text{inv}} := \mathbb{E}_{\mathbf{x}\in\mathcal{X},p=1,...,|G|} \left\|\psi(\mathbf{x}) - D\left(E\left(\psi(g^p\mathbf{x})\right)\right)\right\|_2^2 \tag{2}$$

After observing that a linear encoder may fail to learn to be invariant, we construct our encoder as a two-layer MLP with a ReLU hidden layer, but keep the decoder to one layer. Therefore, while the encoder is expressive enough to learn to be invariant, the canonical activations are still reconstructed as sparse linear combinations of the dictionary vectors. This approach is further motivated by previous work showing that more expressive encoders can lead to more interpretable SAEs [21].

**Activation transformation matrix $\mathbf{M}$.** To map the canonical reconstructions back to their original forms, we need to adapt to the symmetries of the model activations and learn how they transform as the inputs to the model are transformed. While closed-form solutions might be possible for certain cases, they are not practical for arbitrary neural networks. Since many group actions we care about, including rotations, can be represented as linear transformations, we hypothesize that a linear transformation should be able to explain to the transformation in the model's activations. Thus, we optimize $\mathbf{M} \in \mathbb{R}^{d \times d}$ to minimize

$$\mathcal{L}_M := \mathbb{E}_{\mathbf{x}\in\mathcal{X},p=1,...,|G|} \left\|\psi(g^p\mathbf{x}) - \mathbf{M}^p\psi(\mathbf{x})\right\|_2^2. \tag{3}$$

We initialize $\mathbf{M}$ as the identity matrix so that it fits perfectly right away if the model has learned invariant representations, and we optimize it using the Adam optimizer [22].

## 4 Experiments

**Dataset and models.** We create a synthetic image dataset where each image contains four shapes. There are 8 possible shapes, each can be in one of four positions (see Figure 2). Applying $90°$ rotations to the images yields either two or four possible orientations for each shape. We train MLP and CNN-based autoencoders as base models where the task is to compress and reconstruct the image (see Appendix C for training details). Our task is to interpret these autoencoders.

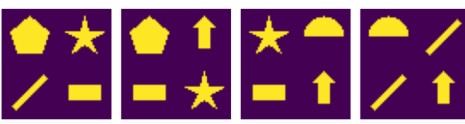

Figure 2: **Sample images from our dataset**. Each image contains one of eight possible shapes in each of its four quadrants.

**Probing Tasks.** We define 180 binary probing tasks to evaluate the downstream usefulness of the features discovered by SAEs (see Appendix C). The tasks are split into four subsets, based on if a shape is in an image (**S**), and in a specific position (**SP**), in a specific orientation (**SO**), and in both a position and an orientation (**SPO**). Note that only the **S** tasks are invariant to rotations. It is desirable

that a small number of latents encode the concepts used to separate the images in the binary probing task. Thus we first identify a small set of latents that maximally differ between the two classes by filtering the SAE latents with the highest absolute difference between the two classes [23]. Then for each task, we train three different probes over the truncated latents as well as the activations, and report results from the best performing probe averaged across all tasks. The probing methods are kNN, logistic regression, and XGBoost, with the XGBoost performing the best overall.

**Training**. Training of our equivariant SAE consists of two independent steps that can be done in any order or in parallel. In the first step, the encoder and decoder weights are updated to minimize $\mathcal{L}_{\text{inv}}$ (Equation 2), and in the second step, the matrix $\mathbf{M}$ is updated to minimize $\mathcal{L}_M$ (Equation 3). Thus the SAE parameters and $\mathbf{M}$ are trained using different objectives, and the training of one does not affect the training of the other. They can also be trained over different datasets, although we use the same dataset for both in our experiments.

**SAE setup and baselines.** We train all SAEs in our experiments over the 256-dimensional middle-layer activations of the base models, and compare our equivariant SAE with two regular TopK SAEs (linear encoder and decoder). The equivariant SAE and the first regular SAE both have an expansion factor of 8, resulting in 2,048 latents. The second regular SAE has $|G|$ times the number of latents, corresponding to learning separate latents for each orientation of equivariant features for a total of 8,192. The regular SAEs are trained by augmenting the dataset with $90°$ rotations.

## 5 Results

RESULT 1: $\mathbf{M}$ **can be learned effectively, indicating that activation-space transformation can be explained by a linear transformation.** Over the middle layer activations of both of our base autoencoders, we optimize $\mathbf{M}$ for 150 epochs and observe that it can be learned with an average ($\pm$ std) $R^2$ of $0.987 \pm 0.001$ between the ground truth and predicted activations across all transformations. As a naive baseline, setting $\mathbf{M} = \mathbf{I}$ results in average $R^2$ values $0.05 \pm 0.08$ and $0.49 \pm 0.03$ for the CNN and MLP autoencoders, respectively.

RESULT 2: **Equivariant SAEs learn more informative directions in activation space, leading to increased performance on group-structured probing tasks.**. Figure 3 shows the classification performance on the 180 tasks of the XGBoost probes with the CNN autoencoder for truncation length 32 and TopK values 8, 16, and 32 (see Appendix D for results with different truncation lengths and the MLP autoencoder). The main outcomes are:

- When probing over the SAE latent activations, the equivariant SAE performs the best, even matching and exceeding the performance of the base model activations in some setups on the **S** group of tasks. Its performance expectedly drops for the equivariant tasks since its latent activations are learned to be invariant. The regular SAE with the two-layer encoder outperforms the SAEs with one-layer encoders, especially with higher $K$. This indicates that it can adapt its latent activations to the symmetries of the data, although at lower sparsity levels.

- Across all tasks and setups, the equivariant SAE outperforms regular and wide SAEs when probes are trained over the truncated reconstructions, and is matched by the two-layer encoder SAE only at low sparsity ($K = 32$).

RESULT 3: **Equivariant SAEs lag behind in the reconstruction/sparsity frontier**. Figure 4 displays sparsity (L1) and the reconstruction (loss when SAE is spliced into the base AE) performance of the SAEs in our experiments. While for small $K$ values the equivariant SAE matches the wide SAE's reconstruction performance, regular SAEs generally have sparser latents. The invariance objective also tends to make the models less sensitive to the choice of $K$, as can be seen from the smaller range of sparsity (L1) values the equivariant SAE results in compared to a regular SAE with a two-layer encoder. Finally, the improvement in reconstruction from the invariant to the equivariant SAE further shows the effect of learning $\mathbf{M}$.

## 6 Conclusion

We have presented early results showing that adding domain-specific properties such as group equivariance to sparse autoencoders can significantly increase their utility in domains beyond language.

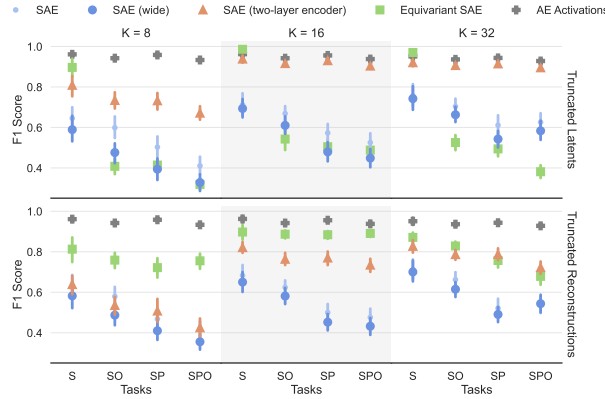

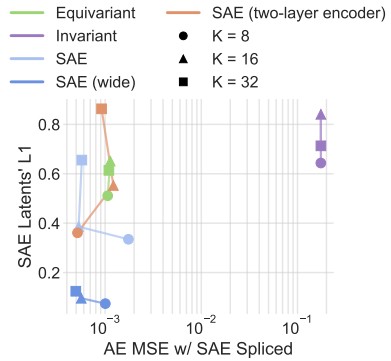

Figure 3: **Latent and reconstruction probing performance**, with truncation length 32 and the CNN autoencoder, comparing a regular SAE (one-layer encoder), an SAE with $|G|$ times the latents (wide), and an SAE with a two-layer encoder with our equivariant SAE. Performance of probes over the base model activations are duplicated for easier comparisons and reflect the upper bound.

Figure 4: **Sparsity/reconstruction performance of SAEs** for varying TopK values. The x axis corresponds to the base autoencoders' reconstruction performance when their intermediate activations are passed through the SAEs, and the y axis denotes the L1 norm of the SAE latent activations.

Our first result is that a single matrix can explain more than 98% of the variance in how the activations of a neural network transform as its inputs are transformed with the action of a discrete cyclic group. We then used this result to design equivariant SAEs that discover features that lead to better probing performance than regular SAEs, indicating that they are more useful in downstream applications.

**Limitations and future work.** Although our results are promising, they are so far limited to a synthetic task, relatively small models, and a small discrete group, and thus many important questions remain to establish the practical usefulness of adaptively equivariant SAEs: Can $\mathbf{M}$ be learned as effectively in larger models such as frontier foundation models? Can the optimization of $\mathbf{M}$ be improved by better utilizing the group structure, e.g. by constraining the optimization to $\mathbf{M}^{|G|} = \mathbf{I}$? Does the two-layer encoder qualitatively alter what kind of features are discovered, and how can the features best be labeled incorporating the knowledge of how they transform with $\mathbf{M}$? Finally, can the reconstruction performance of the equivariant SAE be improved to match that of regular SAEs?

While the concept of interpretability is domain-agnostic, progress in mechanistic interpretability has largely been driven by its applications in language, which has led to certain concepts such as group equivariance being represented far less prominently than they are in the broader ML literature. Our results highlight the potential benefit of bridging that gap and tailoring mechanistic interpretability tools for domains beyond language, despite being early results for a work in progress.

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

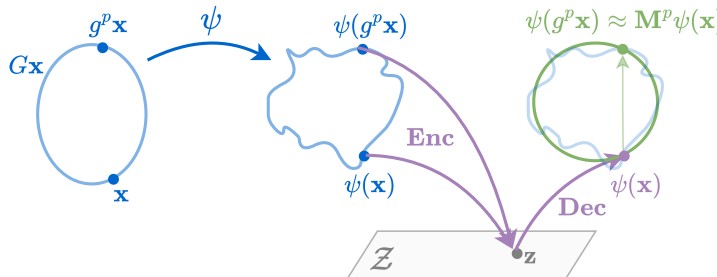

Figure 5: *Left:* A **neural network** $\psi$ transforms an orbit $G\mathbf{x}$ into more complex, non-linear structures. The **invariant SAE** maps all activations in a transformed orbit to the same latent point, and reconstructs a canonical activation. The **matrix M** then transforms the reconstructions back to their original forms, achieving **equivariance** by approximating the transformed orbit in activation space. *Right*: Transforming the learned dictionary $\mathbf{D} \mapsto \mathbf{M}^p\mathbf{D}$ allows us to observe which features discovered by the SAE are **invariant** or **equivariant** with respect to input transformations.

## A  Reproducibility

We make our SAE and probing implementations public at `https://github.com/ege-erdogan/equivariant-sae`. We implement our models from scratch using PyTorch [24], and use the scikit-learn [25] and XGBoost [26] packages for the various probing methods. We utilize the OpenCV package [27] to create our dataset.

## B  Background

### B.1  Sparse autoencoders

Neural networks are known to encode more concepts than they have neurons for, a phenomenon known as *superposition* [11]. This implies that individual neurons are not the right level abstraction to analyze what concepts are involved in a neural network's computation. Sparse autoencoders (SAEs) [28, 29] have shown potential in overcoming this challenge, by decomposing the activations of neural networks into more interpretable components that are used by the model.

An SAE is trained to reconstruct the internal activations $\mathbf{x} \in \mathbb{R}^d$ of a model as sparse linear combinations of interpretable *features*. A typical SAE consists of linear encoder and decoder layers, and an activation $\sigma$:

$$\mathbf{z} = \sigma\left(\mathbf{W}_E\mathbf{x} + \mathbf{b}_E\right) \tag{4}$$
$$\hat{\mathbf{x}} = \mathbf{W}_D z + \mathbf{b}_D \tag{5}$$

with weights $\mathbf{W}_E, \mathbf{W}_D$ and biases $\mathbf{b}_E, \mathbf{b}_D$. The latent $\mathbf{z} \in \mathbb{R}^n$ is higher dimensional than the inputs, and thus the *dictionary vectors* (rows of $\mathbf{W}_D$) is said to form an *overcomplete basis* for the activation space.

To avoid memorization and obtain more clear explanations, the latent $\mathbf{z}$ is pushed to be sparse, with a low L0 norm. Since it is difficult to optimize for a low L0 norm directly, SAEs are often trained with L1 regularization to minimize

$$\mathcal{L} := \|\mathbf{x} - \hat{\mathbf{x}}\|_2^2 + \lambda \|\mathbf{z}\|_1. \tag{6}$$

An alternative approach is to enforce the desired L0 directly with an activation function such as TopK [20, 19] which filters only the $K$ highest activations for each input or BatchTopK [30] which filters over a batch.

The features learned by an SAE can then be interpreted by finding the inputs that maximally activate specific latents or observing how modifying the strength of a latent changes the model outputs, such as in the famous "Golden Gate Claude" [31]. The *geometry* of features learned by SAEs have also been an object of study [32, 33], demonstrating that concepts such as days of the week can be represented in an LLM's activations in particular geometric structures.

## B.2 Symmetries & group equivariance

Data in scientific problems often involve various **symmetries**, transformations that preserve certain properties of the data while some properties can transform along with the symmetries. For example, rotating a molecule, moving it in space, or permuting its identical atoms does not change the molecule's identity, but rotating it can change the orientation of certain vector quantities such as the forces acting on each atom.

The frame of reference we associate with such data, e.g. the particular 3D coordinates we assign to atoms in a molecule, is not an inherent property of the molecule or the world but an artifact of our observational bias. Thus to model physical phenomena more faithfully, we would prefer to be independent of particular reference frames, and developing such tools has become an active research area [34].

Symmetries share certain properties such as being composable (subsequent rotations can also be modeled as a single rotation) and invertible (any rotation can be inverted). Moreover, the identity transform is a trivial symmetry for any object, and the order of composition of three transformations does not change the end result. These notions are unified by the definition of a **group** that characterizes symmetry transformations:

**Definition 1** (Group). *A **group** $(G, *)$ is a set $G$ along with an operation $* : G \times G \to G$ such that the following axioms are satisfied:*

- *(Associativity) For all $g, h, j \in G$, it holds that $(g * h) * j = g * (h * j)$.*

- *(Identity element) There exists $e \in G$ such that $e * g = g$ for all $g \in G$.*

- *(Inverses) Any $g \in G$ has an inverse $g^{-1} \in G$ such that $g * g^{-1} = e$.*

*A group is further called **abelian** if its group operation is commutative, i.e. $g * h = h * g$ for all $g, h \in G$.*

Groups are often denoted by their set, e.g. as $G$ alone, omitting the operation. Groups can be *discrete*, such as permutation groups $S_n$ of $n$ objects and the cyclic groups $C_n$ corresponding to rotations of an $n$-gon, or *continuous*, such as the group of rotations in $n$-dimensional space, defined as $SO(n) := \{\mathbf{R} \in \mathbb{R}^{n \times n} : \mathbf{R}\mathbf{R}^T = \mathbf{I}, \det \mathbf{R} = 1\}$.

Groups transform sets of objects such as images or molecules via their **actions**:

**Definition 2** (Group action). *A (left) **action** of the group $(G, *)$ on set $X$ is a map $\alpha : G \times X \to X$, denoted*

$$(g, x) \mapsto \alpha(g, x) = g \cdot x,$$

*that satisfies these axioms for all $x \in X$:*

- *$\alpha(e, x) = x$ with $e$ the identity element in $G$.*

- *$\alpha(g, \alpha(h, x)) = \alpha(g * h, x)$ for all $g, h \in G$.*

*A right group action can similarly be defined, and the set $X$ is said to be a G-**set**.*

**Definition 3** (Orbit). *Let $X$ be a G-set. The **orbit** of $x \in X$ is the set of all points in $X$ reachable by transforming $x$ with $G$, denoted*

$$Gx := \{\alpha(g, x) : g \in G\}.$$

Functions mapping between $G$-sets can then be **invariant** or **equivariant** depending on how they behave in each orbit:

**Definition 4** (Invariant and equivariant functions). *For G-sets $X$ and $Y$ with associated actions $\alpha_X, \alpha_Y$, a function $f : X \to Y$ is G-invariant if for all $g \in G, x \in G$,*

$$f(\alpha_X(g, x)) = f(x),$$

*and G-equivariant if*

$$f(\alpha_X(g, x)) = \alpha_Y(g, f(x)).$$

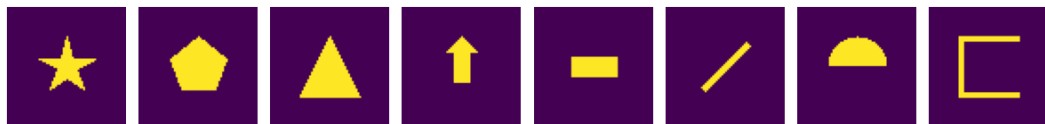

Figure 6: **Set of available shapes in our dataset.** None of the shapes is rotation-invariant, with the horizontal rectangle and diagonal line having two orientations and the other six shapes having four orientations.

Using neural networks, approximate invariance or equivariance can be achieved by augmenting the data with symmetric inputs, or explicitly via additional loss terms [11]. Straightforward ways of achieving exact invariance include limiting the model's inputs to invariant properties of the data, such as internal bond angles in a molecule that are rotation-invariant scalars [35], or averaging the outputs over each orbit [36]. Achieving exact equivariance requires more careful consideration of how the input features are processed in each layer of the neural network, but there exists a wide-ranging literature of equivariant models for various groups and data types [37].

## C   Experimental details

### C.1   Dataset and probing

Figures 6 displays the base shapes in our dataset. When rotated in increments of $90°$, the rectangle and the diagonal shapes have two orientations, and the other six shapes have four. Each image then contains a randomly sampled shape in each of its four quadrants. Precise definitions of our binary probing tasks are then as follows, with a shape's position denoting which of the four quadrants it is in in an image, and its orientation denoting which of the four or two orientations it is in:

- **S**$(s)$: Does the image contain shape $s$ in any position or orientation?
- **SO**$(s, o)$: Does the image contain shape $s$ in orientation $o$ and any position?
- **SP**$(s, p)$: Does the image contain shape $s$ in position $p$ and any orientation?
- **SPO**$(s, p, o)$: Does the image contain shape $s$ in position $p$ and orientation $o$?

This results in a total of 8 **S** (one for each shape), 28 **SO** (2 shapes $\times$ 2 orientations + 6 shapes $\times$ 4 orientations), 32 **SP** (8 shapes $\times$ 4 orientations), and 112 **SPO** (2 shapes $\times$ 2 orientations $\times$ 4 positions + 6 shapes $\times$ 4 orientations $\times$ 4 positions) tasks, for a total of 180 tasks. Note that we report F1 scores rather than raw accuracies as the tasks are not balanced and contain more negative than positive examples, with the share of negative examples ranging from $\sim 58\%$ in the **S** tasks to $87 - 89\%$ in the **SP** and **SO** tasks, and $97\%$ in the **SPO** tasks.

The probe we ultimately report the results from, XGBoost [26], consists of 100 estimators with a maximum depth of 6, and is trained with the learning rate 0.3 and L2 regularization. For completeness, logistic regression probes are trained with learning rate 1e-3 and L2 regularization strength 1e-4, and for the kNN probe we consider the 16 nearest neighbors with respect to Euclidean distance.

### C.2   Base models

We train our base autoencoders for 100 epochs over 10,000 randomly generated samples from our dataset and augmenting with random $90°$ rotations with a batch size of 64 using Adam [22] with learning rate 1e-3. Their architectures are detailed in Table 1.

### C.3   SAEs

The regular SAEs used in our comparisons are typical TopK SAEs with linear encoders and decoders. The equivariant SAEs also have linear decoders and use the TopK activation, but their encoders consists of two linear layers with a ReLU activation in between and hidden dimension of 512. We train our sparse autoencoders for 500 epochs over 10,000 samples from our dataset with batch size 64 using Adam [22] with a learning rate of 1e-3.

Table 1: **Architectures of the MLP and CNN autoencoders.** The first section of each models corresponds to the encoder and the second section to the decoder. We train our SAEs over the pre-activation encoder outputs.

| MLP | CNN |
|---|---|
| Input: 4096 ($64 \times 64$) | Input: $1 \times 64 \times 64$ |
| Linear(4096, 256) 
 ReLU 
 Linear(256, 256) 
 ReLU | Conv2d(1, 16, $3 \times 3$, stride=2, pad=1) 
 ReLU 
 Conv2d(16, 32, $3 \times 3$, stride=2, pad=1) 
 ReLU 
 Conv2d(32, 256, $16 \times 16$) 
 ReLU |
| Linear(256, 256) 
 ReLU 
 Linear(256, 4096) | ConvTranspose2d(256, 32, $16 \times 16$) 
 ReLU 
 ConvTranspose2d(32, 16, $3 \times 3$, stride=2, pad=1, out_pad=1) 
 ReLU 
 ConvTranspose2d(16, 1, $3 \times 3$, stride=2, pad=1, out_pad=1) |

## D    Further probing results

Figure 7 displays further probing results with TopK values 8 and 32. Results generally agree with those presented in Section 4. When probing over the truncated reconstructions, the invariant/equivariant SAEs result in the most accurate probes over all tasks while the performance of the latent activation probes drop with the equivariant tasks as expected.

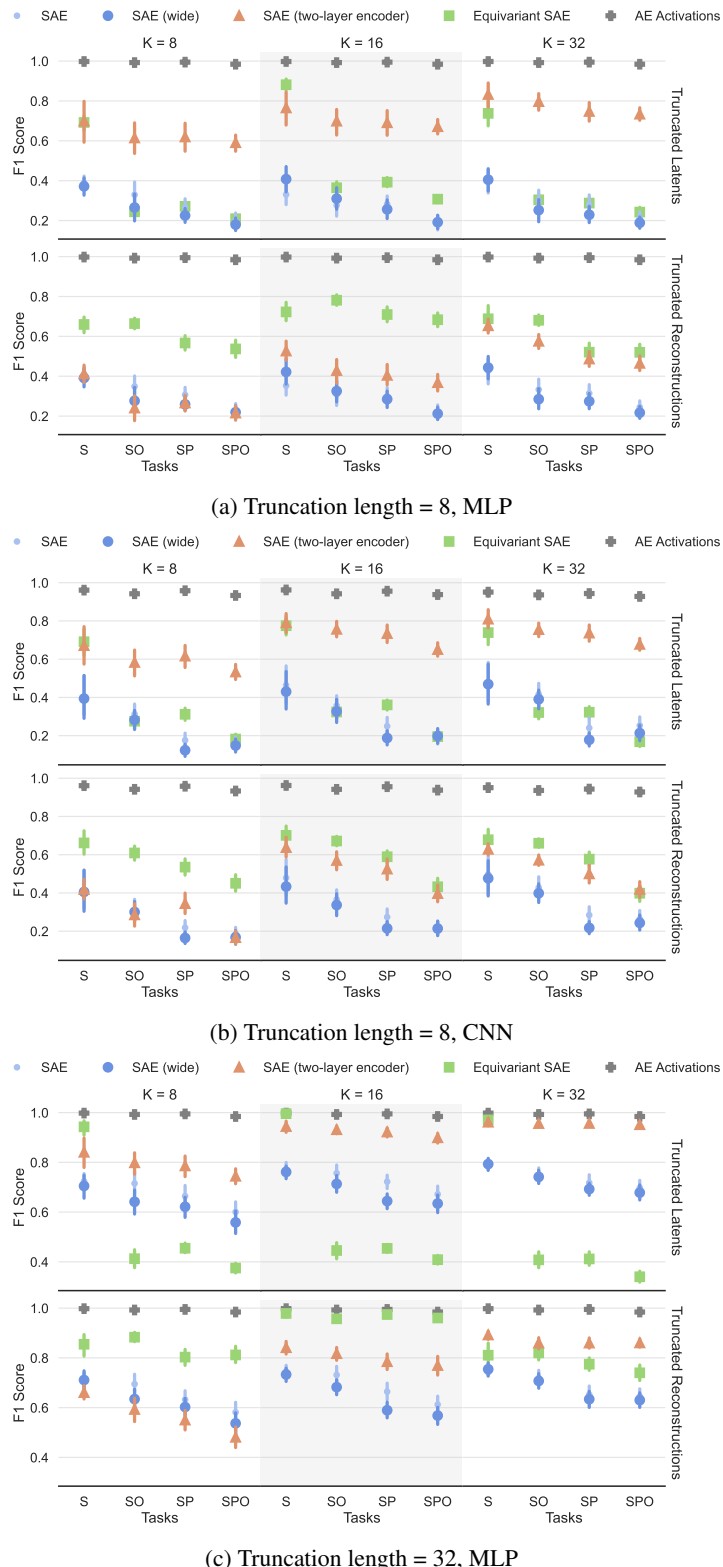

(a) Truncation length = 8, MLP

(b) Truncation length = 8, CNN

(c) Truncation length = 32, MLP

Figure 7: **Further probing results with different TopK values.** Although increasing TopK and the truncation length increases the performance of the probes trained over the regular SAEs activations and reconstructions, results follow a similar trend with those in Figure 3, with the equivariant SAE leading to the best overall probes over its reconstructions.

