# OpenReview forum: "Group Equivariance Meets Mechanistic Interpretability: Equivariant Sparse Autoencoders"
_NeurIPS.cc/2025/Workshop/UniReps — UniReps2025_

### Official Review · Reviewer_CyEc · 2025-09-11
**Combining two useful ideas, but poorly formulated. Preliminary results are provided, but not on the mechanistic interpretability.**

**Confidence:** 4

**Review:**

SAEs have gained popularity in mechanistic interpretability of large models. Started with language, now widly being applied into vision models, and in science (e.g., proteins models).


Pro:

1. This paper is very timely to incorporate equivariance into SAEs for mechanistic interpretability of models trained on scientific data.

2. They propose a simple approach to learn a linear group transformation (applicable to learning transformation such as rotations, tralsation, ...).  This emphasizes the importance of learning the group transformation which is not known/trivial on the input activation and also importance of approximate symmetries.

3. Clear contribution section.


Cons:

1. The main feature of SAEs trained on activations of large models is to extract learned concepts by the model (a.k.a mechanistic interpretbaility). However, this paper evaluates their SAEs for some downstream tasks without visualization of the learned concepts, and a discussion on the impact of equivariance on learned concepts.


2. I found the formulation weak with not-well explained choices. The formualtion is not well-written.
	- There is no non-linearity between E and D (see eq. 1, 2). It's not clear why there is no nonlinearity between E and D.
	- Two linear encoder one after another are still linear encoder. Is there a non-linearity there?
	- If there is no non-linearity then how the SAE is imposing sparsity.

3. On results: SAE community is moving beyond rec error. MSE is not a useful metric in evaluating the performance of SAEs. One may give low error; what makes SAEs apart from one another is their ability to extract attributes with different characteristics. I suggest not to emphasize on rec loss and explore more the characteristics of the concepts. For example, visualization of the decoder concepts that are invariant and those that are equivariant to the transformation (a property that is emphazies in fig1 but not explored in the results).

minor:

- first sentence, provide citations for language.

- the bold text is bothering. I stonrgly suggest not to bold pharases in the middle of a sentence.

- the "adaptive" part of formulation while mentioned in the abstarct and intro was not discussed (I missed this point) in the main part of the paper. Did you mean that you ae learning M?

- They study two models of MLP and CNN. It would be useful to look into the difference between the two.

- I suggest to also add analysis with L0. Topk extract sparsity based on L0, not L1.

**Score:**

3

**Topic Fit:**

2

---

### Official Review · Reviewer_ZnBL · 2025-09-13
**Equivariant-invariant sparse autoencoders - promising results on synthetic datasets**

**Confidence:** 4

**Review:**

This work incorporates group symmetries into the SAEs to develop equivariant SAEs. Equivariant SAEs are made by an invariant autoencoder, which is made equivariant with the addition of an additional matrix $\mathbf{M}$ to estimate how the activation transforms when given as input an image and its affinely transformed version.

Results are promising, especially for the probing tasks. However, it's not clear:
- if the total loss is the sum of $\mathcal{L}_\textrm{inv}$ and $\mathcal{L}_M$
- What is the base autoencoder (Figure 4, for example)? Is it the regular/standard SAEs?
- Why have two hidden layers in the encoder of the equivariant SAE?

**Score:**

3

**Topic Fit:**

2

---

### Official Review · Reviewer_NUzJ · 2025-09-15
**This work proposes a new architecture for learning invariances, equivariances using SAEs for feature learning for downstream tasks**

**Confidence:** 4

**Review:**

The architecture proposed for learning the invariances and equivariances using SAEs in novel. The authors mention that these adaptive equivariant SAEs learn features that are useful for downstream tasks.

While the architecture is interesting, the symmetries seems to be learned through the matrix M, and only linear symmetries. Though the authors show that it works for synthetic datasets involving rotation, they do not provide any directions of how this would be handled for non-linear symmetries. There have been earlier attempts at inferring linear symmetries using the theory of Lie algebra and Lie groups given below. How does this approach compare with existing approaches?

Yang, Jianke, et al. "Generative adversarial symmetry discovery." International conference on machine learning. PMLR, 2023.

There has also been work on extending this to non-linear symmetries given below.

Yang, Jianke, et al. "Latent space symmetry discovery." arXiv preprint arXiv:2310.00105 (2023).

The authors seems to indicate the use of SAEs to for mechanistic interpretability. However, recent research in interpretability of SAEs applied to LLMs show limited use.

Leask, Patrick, et al. "Sparse autoencoders do not find canonical units of analysis." arXiv preprint arXiv:2502.04878 (2025).

The architecture proposed in this paper is interesting. But, as indicated above there has been work on inferring the linear and non-linear symmetries. It would be nice if this work could compare with the earlier methods. The results using synthetic data are limited. More experiments with standard datasets for symmetry discovery could help. The interpretability claims using SAEs are not well established

**Score:**

2

**Topic Fit:**

2

---

### Official Review · Reviewer_M2f8 · 2025-09-16
**Synthetic data but Adaptively Equivariant SAEs give promising results**

**Confidence:** 3

**Review:**

**Summary:** The paper presents a novel contribution called "Adaptively Equivariant SAEs", expanding the mechanistic interpretability field. The authors show how a single matrix can explain how activations change when (synthetically created) images are rotated, by incorporating symmetries between data groups.

**Strengths:**
- Novel ideas and insights. A single matrix explanation of activation is very interesting and could have wide applications.
- The paper overall is well structured and easy to follow. The simple example about what constitutes an invariant feature is very helpful. So are the appendices.
- Symmetries are fundamental to biological problems like proteins, DNA as well. This paper tries to bridge the gap between equivariance and mech interpretability. This is a well motivated problem.

**Weaknesses:**
- Limited experimental validation - The authors test only a small synthetic 2D dataset with basic figures and augmentations of 90 degrees. While it is a good initial proof-of-concept, there is no real evidence that this method will hold up in real-world scientific datasets. They do acknowledge this in their limitations section, however.
- Some baseline information on how another model would perform with and without the same data augmentations would be useful to provide the readers more context.
- The method as mentioned already, performs poorly on reconstruction compared to regular SAEs. From Fig 3, the equivariant SAE underperforms vs regular SAEs across all Ks. Although the initial results are exciting, this makes me question the overall utility in large datasets, and more complex models.
- Details on compute - time, memory, cost etc in comparison to regular SAEs would be a good addition to help understand if using one or the other is better in certain situations. This raises practical trade off considerations, even if the performance is superior in the real world.

**Score:**

4

**Topic Fit:**

3